# Development of a Wearable Haptic Glove Presenting Haptic Sensation by Electrical Stimulation

**DOI:** 10.3390/s23010431

**Published:** 2022-12-30

**Authors:** Dongbo Zhou, Wataru Hayakawa, Yoshikazu Nakajima, Kotaro Tadano

**Affiliations:** 1Institute of Innovation Research, Tokyo Institute of Technology, Yokohama 226-8503, Kanagawa, Japan; 2Department of Biomedical Informatics, Institute of Biomaterials and Bioengineering, Tokyo Medical and Dental University, Tokyo 101-0062, Japan

**Keywords:** wearable haptic device, electrical stimulation, haptic sensation, stimulus signal, stimulation area, design optimization

## Abstract

Most haptic devices generate haptic sensation using mechanical actuators. However, the workload and limited workspace handicap the operator from operating freely. Electrical stimulation is an alternative approach to generate haptic sensations without using mechanical actuators. The light weight of the electrodes adhering to the body brings no limitations to free motion. Because a real haptic sensation consists of feelings from several areas, mounting the electrodes to several different body areas can make the sensations more realistic. However, simultaneously stimulating multiple electrodes may result in “noise” sensations. Moreover, the operators may feel tingling because of unstable stimulus signals when using the dry electrodes to help develop an easily mounted haptic device using electrical stimulation. In this study, we first determine the appropriate stimulation areas and stimulus signals to generate a real touch sensation on the forearm. Then, we propose a circuit design guideline for generating stable electrical stimulus signals using a voltage divider resistor. Finally, based on the aforementioned results, we develop a wearable haptic glove prototype. This haptic glove allows the user to experience the haptic sensations of touching objects with five different degrees of stiffness.

## 1. Introduction

Presenting haptic sensations to the operator can improve operational precision, efficiency, and presence realism when implementing virtual reality, augmented reality, robot-aided surgery, and teleoperation [1]. The device for providing haptic feedback is classified into two types based on its mechanical grounding configuration: grounded and ungrounded. Touch X [2], Omega 3 [3], and SPIDER [4] are examples of grounded devices with their bases fixed to the ground. These devices use the torques of the internal motors to generate haptic sensations. However, the fixed base limited the movement range and the operational motion. The ungrounded type haptic devices do not limit the operator’s range of motion because they have no fixed bases. However, hand-held devices, such as the operation end of Meta Quest and Nintendo Switch, cannot provide force feedback. The glove-like devices, such as HaptX Glove [5] and CyberGrasp [6], can provide force feedback, but their weight increases the operator’s workload. To realize the lightweight and wearable ungrounded haptic device being able to produce both tactile and force feedback. Terrile et al. employed the lightweight spring made of shape memory alloy to produce force feedback at a glove-like device [7], but the response time might be an issue. Ji et al. developed an untethered finger-mounted gadget using dielectric elastomer actuators [8]. This gadget can generate force sensation at the fingertips by wrapping them, but it is challenging to wrap the entire arm in order to generate force sensation at the arm.

To address this issue, many studies developed new approaches for generating haptic sensations via vibration [7], ultrasonic wave [8], air pressure [9], laser [10], magnetic field [11], etc., but these devices lack force generation.

Electrical stimulation, which applies the electrical stimulus signals to the electrodes attached to the human body, is widely used in the treatment and rehabilitation of physical paralysis [9]. When implementing electrical stimulation, electrotactile stimulation current can elicit nerve firing at the subdermal receptors, giving users the sensation of “touching.” Many studies developed haptic interfaces using electrical stimulation. Regarding tactile sensations, Kajimoto proposed the tactile primary color approach, by which various tactile sensations on the fingertip can be generated by choosing the stimulating receptors by different electrical signals [10] and developed a fingertip electrotactile display [11]. Yem et al. developed a finger-mounted haptic display using both electrical and mechanical stimulations [12]. Regarding force sensations, the electrical stimulus signal elicits nerve firing and activity in the antagonistic muscle. Hence, the users feel external forces. Lopes et al. used electrodes that stimulate the arm muscle to provide force sensation when playing a mobile game [13]. Hasegawa et al. proposed a bilateral control technique by which the interaction force can be presented using functional electrical stimulation at the shoulder and elbow joints [14]. Mizuhara et al. used electrical stimulation to present the sting sensation as snake biting or injection [15]. Because the electrodes adhering to the body are lightweight, the operator’s free motion is not limited. Electrical stimulation is a promising alternative approach to presenting haptic sensations in a lightweight manner.

However, current haptic devices using electrical stimulations, such as those introduced in [13,15,16,17], combine electrical and mechanical stimulations. However, the mixed modalities of stimuli increase the weight and complexity of the whole device. To the best of the authors’ knowledge, no existing study has provided a haptic device that solely uses electrical stimulations to present both tactile and force sensations.

The first reason is the “noise” sensation when simultaneously stimulating multiple areas. People’s “contact” sensation on the arm may consist of the activations of multiple muscles, such as the “shake” sensation (haptic), force sensation, and “move” (kinesthetic) sensation at the hand, wrist, and elbow, respectively. Mounting the electrodes to multiple arm areas is expected to produce real “contact” sensation. However, a tradeoff between the stimulation areas and sensation reality exists. When simultaneously stimulating multiple areas, people are prone to sense the “noise” chattering sensation accompanied by the electrical stimulation [14]. The appropriate stimulation areas, as well as the appropriate stimulus signals applied to the electrodes to balance reality and “noise” sensation, have not been determined.

The second reason is the tingling sensation caused by the instability of the stimulus signals. For ease of putting on and removing, wearable haptic devices are typically using dry electrodes. However, when utilizing dry electrodes, the activity of the sweat glands may short-circuit the resistance of the skin. The threshold of tingling may be exceeded by the abrupt rise in stimulus current [18]. The amplitude of the stimulus signal may surpass the threshold, causing uncomfortable tingling sensations. The unpredictability is the main factor reducing the sensation reality when the user receives electrically stimulated haptic sensations [19].

Therefore, this study aims to address these two issues and to develop a lightweight, wearable haptic device that provides electrically stimulated haptic sensation on the forearm. To this end, we first clarify the appropriate stimulation areas and stimulus signals through the subject experiment. Second, we propose a circuit design guideline to generate stable electrical stimulus signals when using dry electrodes. Then, based on the findings, we develop a lightweight and wearable haptic glove prototype and verify its performance.

The remainder of this article is organized as follows. In Section 2, we introduce the subject experiment and the voltage divider circuit guideline, and the developed haptic glove. In Section 3, we describe the experiment verifying the performance of the developed haptic glove. In Section 4, we analyze the findings. In Section 5, we conclude this study.

## 2. Materials and Methods

Regarding the first issue, “tradeoff between the stimulation areas and sensation reality”, we tried a subject experiment in which various combinations of stimulation areas and stimulus signals were applied to human subjects and determined the one that provided the highest haptic reality ranked by the subjects. Regarding the second issue, “tingling sensation caused by the instability of the stimulus signals”, we used the voltage divider circuit to stabilize the amplitude of electrical stimulus when using the dry electrodes and proposed the design guideline for choosing the voltage divider resistor. Based on the results of the two previous experiments, we developed a lightweight and wearable haptic glove prototype.

### 2.1. Propriate Stimulation Areas and Stimulus Signals

#### 2.1.1. Stimulation Areas

According to the principle of force sensation, when a person’s forearm comes in contact with an object, the contact force extends or flexes the muscles at the wrist or elbow; the person actively outputs an opposite movement trend to balance the contact force and thus activates the sensory receptors in the muscles. Therefore, an electrically stimulated force sensation can be generated by applying an electrical stimulus to the elbow biceps and wrist extensor digitorum muscles, which are responsible for elbow and wrist motions, respectively.

Moreover, the shake resulting from the contact activates the subdermal receptors at the palm. Hence, we place gel pad electrodes on three areas of the forearm, as shown in Figure 1.

#### 2.1.2. Subjects and Experimental Devices

We enrolled eight subjects, all of whom were men aged 22–35 years. None of the subjects had any prior experience with haptic devices that use electrical stimulation.

The device that generated the stimulation current was a general-purpose stimulus generator (STG4008, multichannelsystems Co., Reutlingen, Germany). The electrodes used in this experiment are gel pad electrodes (Setsu Planning Co., Ltd., Fukuoka, Japan). The electrodes stimulating the elbow biceps and wrist extensor digitorum muscles were square (50 × 50 mm), and the electrodes stimulating the hand palm were circular (diameter: 30 mm).

#### 2.1.3. Stimulus Signals

In this study, the stimulus signals applied to the electrodes were bidirectional square waves of current. The general process of contacting an object consists of three stages: impact, touch, and release. The features of the interaction force in the three stages are a high abrupt force with a short duration, a medium stable force with a long duration, and decreasing force with a short duration. Hence, we imitate the three stages by applying a bidirectional square wave of current with three different amplitudes (Figure 2a) to the electrodes.

Regarding tactile sensation, the Meissner corpuscles (RAI) receptors under the skin were responsible for sensing shake [20]. Hence, we imitate the firing rate of RAI receptors by applying intermittent bidirectional square waves (Figure 2b, lower) to the palm electrode as the stimulus signals. Merkel cells are another haptic receptor under the skin that continuously fires when pressed. As a result, we also tried the continuous stimulus signal, as shown in the lower Figure 2b. The frequency of all stimulus signals in Figure 2 is 50 Hz, and the pulse width is 200 µs, these levels of stimulus were used in other studies using electrical stimulation to generate haptic sensations [14,17,21,22]. Before the experiment, we investigated the magnitudes of each stimulus signal presenting the most real sensation for each subject. Table 1 lists the stimulus amplitude parameters applied to eight subjects.

Regarding the bidirectional square wave with changing amplitudes applied to the wrist biceps and elbow extensor digitorum muscles, the amplitude is $A_{\mathrm{impact}}$ at the first 0.06 s of contact, $A_{\mathrm{touch}}$ when the hand palm remains in contact, and $A_{\min}$ in 0.12 s after the hand leaves the ball. Regarding the continuous or intermittent stimulus signals applied to the hand palm, the $A_{\mathrm{con}}$ and $A_{\mathrm{int}}$ in Table 1 were used.

#### 2.1.4. Experimental Conditions

Table 2 lists the conditions applied in the subject experiment. Figure 3 shows the appearance of the experiment. The subject sat in front of a display and moved their hand back and forth at a normal speed. Leapmotion (LM-010, Ultraleap Co., Bristol, UK), an optical hand tracking module, measured the position of the hand and transformed the hand position to that in the virtual world, in which a virtual ball exists. When the subject hand in the virtual world invaded the range of the ball, a stimulus current was generated and applied to the electrodes.

At each condition, the subject rated the reality of the “contact” sensation generated by the electrical stimulation, between a scale of 1 (unreal) to 5 (real), as well as the arm area where the haptic sensation was sensed.

#### 2.1.5. Propriate Stimulation Areas and Stimulus Signals

Figure 4a shows the reality rated by the subject after receiving the stimulated haptic sensation applying the conditions in Table 2. This result shows that the H + W condition (stimulating the hand palm and wrist extensor digitorum muscle and applying an intermittent stimulus to the palm) could generate the haptic sensation of “contact” with the highest reality and lowest dispersion compared with other conditions.

Figure 4b shows the proportions of the area where the haptic sensation was sensed, as reported by the subjects from the seven options represented by different colors. Because the subject made contact with the virtual ball by hand, the correct answer should be “Palm” or “wrist front.” Figure 4b shows that the H + W condition can provide the highest proportion of the correct answer, except the H condition (only stimulates the hand palm), in which the rank of reality is low. Hence, using the combination of the stimulation areas and stimulus signals of condition H + W (stimulating palm with intermittent signals and stimulating wrist extensor digitorum muscle with change amplitude signals) may be appropriate for providing a “contact” haptic sensation at the forearm.

### 2.2. Eliminating Tingling Sensation

#### 2.2.1. Voltage Divider Circuit Design Guideline

The presentation of haptic sensations on the fingers is also necessary for sensation reality. Wearable haptic devices are often used with dry electrodes in small areas such as fingers for ease of putting on and removing. Figure 5 shows the circuit schematic used in this study. The resistance of finger skin at the dry condition ($R_{\mathrm{dry}}$) is 2–5 kΩ. However, $R_{\mathrm{dry}}$ may be short-circuited due to sweat. Hence, the stimulus current may increase and exceed the tingling sensation threshold. Connecting a resistor with a high resistance value in series (voltage divider resistor) can constrain the change in the stimulus current. A guideline for determining the voltage divider resistor is required.

According to the Kirchhoff theorem, the maximum transient stimulus current $I_{\mathrm{tran}}$ applied on the finger when the skin resistance becomes zero can be calculated using the following function.
(1)$$I_{\mathrm{tran}}=\left(1+\frac{R_{\mathrm{dry}}}{R_{\mathrm{div}}}\right)I_{\mathrm{ref}}$$

In Equation (1), $I_{\mathrm{ref}}$ is the stimulus current before resistance change. According to our pilot study, the tingling sensation appears when the stimulus current is approximately 3.0 mA. Hence, considering a safe coefficient of 0.8, $I_{\mathrm{tran}}$ should be <2.4 mA, and the voltage divider resistor $R_{\mathrm{div}}$ should satisfy the following function.
(2)$$R_{\mathrm{div}}>\frac{2.4\;R_{\mathrm{dry}}}{2.4-I_{\mathrm{ref}}}$$

Because of the power limitation of the stimulus generator, an extremely large *R* may decrease the stimulus current enough to generate a real sensation. Hence, $R_{\mathrm{div}}$ should also satisfy.
(3)$$R_{\mathrm{div}}<\frac{V_{\max}}{I_{\mathrm{ref}}}-R_{\mathrm{dry}}$$

Equations (2) and (3) can be used to choose the voltage divider resistor.

#### 2.2.2. Verification Experiment

In this study, we used the dry electrodes (EL503, Biopac Systems, Goleta, CA, USA) shown in Figure 6 at the finger area, of which the diameter is 9 mm and the contact area with the finger skin is 50 mm^2^ (the center of the electrode does not make contact with the skin, those areas are not counted as contact areas). The eight subjects mentioned in Section 2.1 also participated in this experiment. Table 3 lists the parameters of the stimulus signals in the verification experiment. We applied the stimulus 500 times to each finger on the right hand using the dry electrodes. The subjects were asked whether they felt a tingling sensation after each application of the stimulus.

We determined a voltage divider resistor of 22 kΩ using Equations (2) and (3). The number of times that the subject experienced a tingling sensation was zero. The voltage divider circuit designed following the proposed guideline effectively reduced the occurrence probability of the tingling sensation.

### 2.3. Wearable Haptic Glove System Prototype

On the basis of the findings in Section 2.1 and Section 2.2, we developed a wearable haptic glove system prototype, as shown in Figure 7. This system consists of a haptic glove with electrodes, a stimulus generator that generates stimulus signals, and a head mount display (HMD, Oculus Quest2, Meta Platforms, Inc., Menlo Park, CA, USA) with a camera that tracks the position of the user’s arm and converts the hand position into that in the virtual world. If the user’s hand touches an object in the virtual world, the computer calculates the interaction force using the game engine Unity (Unity Software Inc., San Francisco, CA, USA) and tells the stimulus what kind of stimulus signals to generate.

On the basis of the findings in Section 2.1, when using the developed haptic device, the stimulation areas are the fingertip, the palm and the wrist extensor digitorum muscle. The haptic glove consists of a glove and an elbow band. The total weight of the glove and elbow band is 65.1 g. Figure 8 shows the electrode layout inside the haptic glove. When the user wears the elbow band, two gel pad electrodes inside the black elbow band will contact the wrist extensor digitorum muscle. The dry electrodes in the white glove can be found at the five fingers and the upper/lower palm. We employed dry electrodes inside the hand glove for easy wearing and removal. The elasticity of the elbow band and glove can maintain certain contact forces between the skin and electrodes, allowing adherence to the corresponding stimulation areas to be maintained without sliding.

## 3. Results

We verified the performance of the developed wearable haptic glove system by testifying whether it can successfully provide users with haptic sensations of different stiffness. Figure 9 shows the appearance of the experiment, with the subject wearing the HMD and haptic glove. The view of the subject is the virtual world. Five virtual balls with numbers one to five are lined up. The five balls are identical in size and appearance, but the stiffness was varied.

### 3.1. Stimulus Signals

The glove part is used to generate the tactile sensation. The experimental result discussed in Section 2.1 shows that the intermittent bidirectional square waves are the appropriate stimulus for tactile sensation. Therefore, we used the intermittent bidirectional square waves shown in Figure 2b. However, the dry electrodes inside the glove part are different from the gel pad electrodes used in the experiment in Section 2.1. As a result, the amplitude of the bidirectional square waves is 2.4 mA, and the tingling sensation does not appear.

The elbow band part is used to generate force sensations. The force sensation of contacts is generated by the contraction of muscles, and other studies verified that the contraction force is proportional to the stimulus [23]. Therefore, we present the sensations of various stiffness levels to the subject by generating different levels of muscle contraction, which is realized by varying the amplitude of the stimulus. Figure 10 shows the waveform of the stimulus represented by Equation (4).
(4)$$A=\left\{\begin{array}{cc} A_{\max} & \left(l<r-d\right)\\ \frac{A_{\max}-A_{\min}}{d}(l-r)+A_{\min} & \left(r-d<l<r\right)\\ 0 & \left(r<l\right) \end{array}\right.$$

In Equation (4), $A_{\min}$ is the stimulus amplitude in which the subject’s wrist extensor digitorum muscle begins to contract, and $A_{\max}$ is the stimulus amplitude in which muscle contraction can output a force of 5 N (considering the output limitation of the stimulus generator). The method of obtaining $A_{\min}$ and $A_{\max}$ is described in Appendix A. r is the radius of the virtual ball; l is the distance between the subject’s virtual hand palm and the ball center; when the subject’s hand palm invades into the virtual ball by a depth of d, the current amplitude becomes $A_{\max}$. Herein, we set five levels of d, which are $\frac{1}{5}r$ to $\frac{5}{5}r$, at a step of $\frac{1}{5}r$. Hence, the five levels of the coefficient $\frac{A_{\max}-A_{\min}}{d}$ values in Equation (4), work as the five stiffness levels.

### 3.2. Experimental Procedure

Throughout the experiment, the subject sat and wore the HMD. After the distance between the virtual hand and the virtual ball in the virtual world is calibrated. The subject moved forward with the virtual hand. When the virtual hand palm made contact with the virtual ball, the haptic glove would present a contact haptic sensation. In one trial, the subject is instructed to touch all five virtual balls and rank their stiffness. We enrolled eight subjects, and each subject repeated the stiffness ranking three times on different days. The five stiffness levels are randomly distributed to the five virtual balls; the subject cannot predict or remember the stiffness order before each trial.

### 3.3. Experimental Results

Figure 11 shows the proportion of the correct ranking of each stiffness level. (The number 1 to 5 in Figure 11 is the order of the stiffness level, not the number shown on the right side of Figure 9.) The proportion of correctly ordering each stiffness level (beyond 70%) is greater than that of the wrong answers (below 20%). This result demonstrates that the subject can correctly distinguish the difference in the stiffness of the virtual object from the haptic sensation provided by the haptic glove.

## 4. Discussion

Regarding the appropriate stimulation area, when the elbow biceps muscle stimulation is added, the area where the “contact” was sensed tends to drift to the back of the forearm (the light gray component), as shown in Figure 4b. The feedback from the subject shows that the sensation is more like the forearm being pulled than pushed. The sensation generated from the contraction of the elbow biceps muscle overwhelmed the other sensations. Appendix A shows that only stimulating the wrist extensor digitorum muscle can produce an output force of 5 N. Thus, when generating the haptic sensation of light contact, it is unnecessary to stimulate the elbow biceps muscle.

In this study, we stimulated the wrist extensor digitorum and the elbow biceps muscles but did not stimulate muscles controlling the shoulder. The reason for this is that the motor nerves of the infraspinatus and subscapularis (muscles responsible for the motion of the shoulder area) are deep. In the pilot study, we attached the electrodes to the shoulder area. However, the muscles did not contract even when the current amplitude reached the maximum of the stimulus generator. Moreover, according to the subject, the chattering sensation in the shoulder area was obviously unreal. However, in real life, when the impact is strong, the shoulder area also provides some of the impact sensations. The concluded appropriate stimulation areas are for the light contact sensation at the forearm.

Regarding the stimulation pattern, we used electrical muscle stimulation (EMS) that stimulates the motor nerve of the muscle. The “noise” sensation is from muscle activity. Some other studies used tendon electrical stimulation (TES) [17,24] that stimulates the sensory nerves because TES can generate a force sensation without causing muscle contractions, resulting in less “noise” sensation than using EMS. However, TES cannot generate the impact sensation, and the individual variation in the received sensation is significant.

Regarding the stimulus signals, we applied the bidirectional square waves to all electrodes in the haptic glove. The reason for using bidirectional waves is that the single-direction waveform generates a current in only one direction; the polarization phenomenon between the skin and electrode tends to cause skin inflammation and skin damage. We also tried triangle and sine waves, and the generated haptic sensations did not differ from that of the square wave.

The frequency of the stimulus signals used in this study was 50 Hz. Although applying a high-frequency stimulus can reduce the “noise” chattering sensation, a stimulus with a frequency >1 kHz cannot cause muscle contraction and thus cannot generate a force sensation. In the pilot study, we tried a 200-Hz stimulus, but the wrist extensor digitorum muscle (responsible for wrist motion) did not contract, even when the amplitude reached the maximum output of the stimulus generator (16 mA). As a result, we compared the stimulus signals used in other similar studies and arrived at the frequency of 50 Hz used in this study.

The voltage divider circuit is simple and effective for preventing tingling sensations. The author also tested the circuit without the voltage divider resistor and applied moisturizing cream on the finger, which is the traditional method of maintaining stable skin resistance. The subject feels a tingling sensation every 5–10 stimuli. For future applications, a larger-scale verification experiment with more subjects and trials is still needed. Another study developed real-time impedance feedback to stabilize the electrotactile stimulation [11]. However, the system circuit is complicated.

Regarding the experimental result of presenting the haptic sensation of different results, although the correct rate overwhelms the incorrect rate, the proportion of incorrect ordering increases when the reference stiffness level is high (levels 4 and 5). The reason is that the just noticeable difference to distinguish difference increases as the reference increases (Weber–Fechner law). When the reference stiffness level is high, the magnitude of the change to the reference stiffness level should be larger to make the subject identify the difference.

The generated sensation across the subjects when applying stimulus signals varies greatly because of the various thicknesses of skin and fat. In the future, the quantitative relationship between the reference stiffness and sensed stiffness will be clarified. The stimulus signals applied to the electrodes inside the glove part that generates the tactile sensation are the same for each subject. However, the sensitivity of each finger may differ. It is necessary to define the sensitivity and appropriate stimulus amplitude for each finger.

## 5. Conclusions

In this study, we developed a wearable haptic glove prototype that uses electrical stimulation to present “contact” sensations on the user’s forearm. This haptic glove is lightweight and easy to put on and remove. During the development of this haptic glove, we clarified the appropriate stimulation areas and stimulus signals, resulting in high reality and the correct sensing position when the user received the electrically stimulated haptic sensation. Furthermore, we proposed a design guideline for using a voltage divider circuit to prevent “tingling” sensations when adhering the dry electrodes to the body part with a small contact area, such as the finger. The developed wearable haptic glove system can present the haptic sensation of touching objects with different stiffness.

In future studies, we will improve this prototype by addressing the issues mentioned in Section 4, eliminating the wires and cables to the microcontroller and battery, and developing a dataset of the stimulus signals for presenting different haptic sensations. Additionally, we will compare the performances of devices using only electrical stimulations and those that employed the most recent lightweight motors.

## Figures and Tables

**Figure 1 sensors-23-00431-f001:**
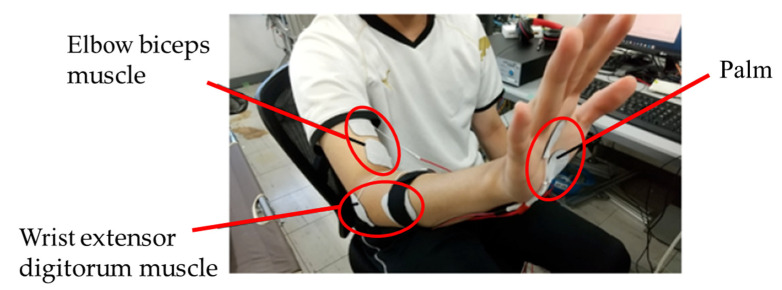
The candidate stimulation areas.

**Figure 2 sensors-23-00431-f002:**
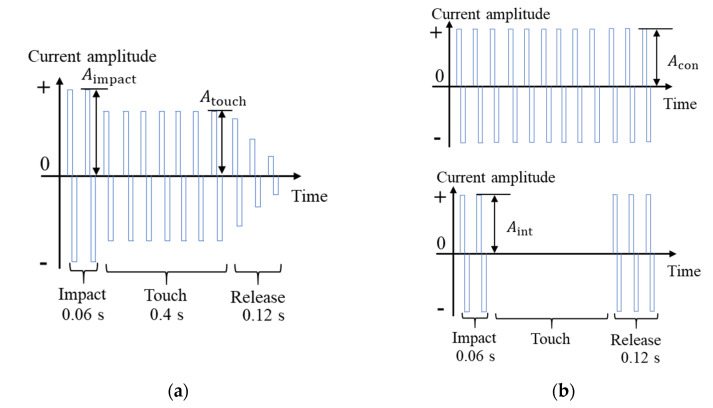
Stimulus signals. (**a**) Bidirectional square wave with three kinds of amplitudes and (**b**) continuous and intermittent bidirectional square waves.

**Figure 3 sensors-23-00431-f003:**
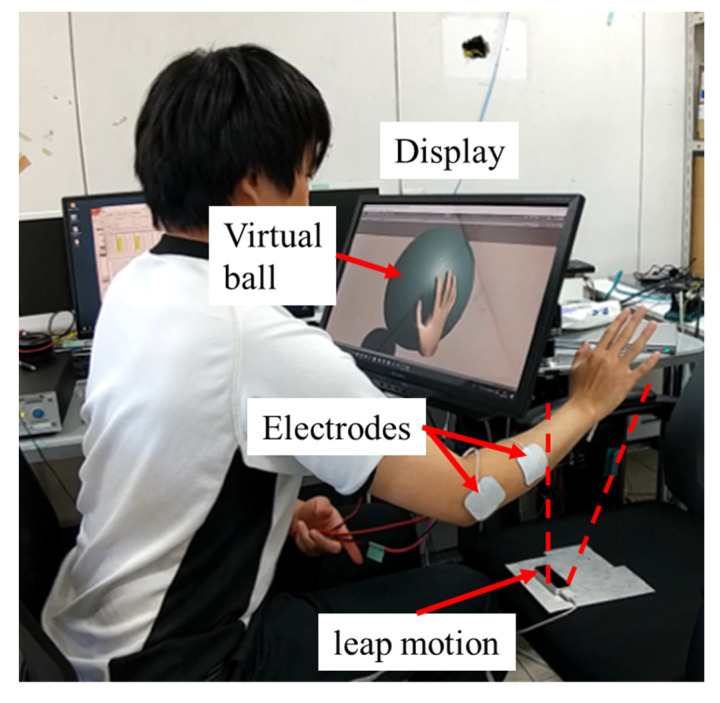
Appearance of the experiment.

**Figure 4 sensors-23-00431-f004:**
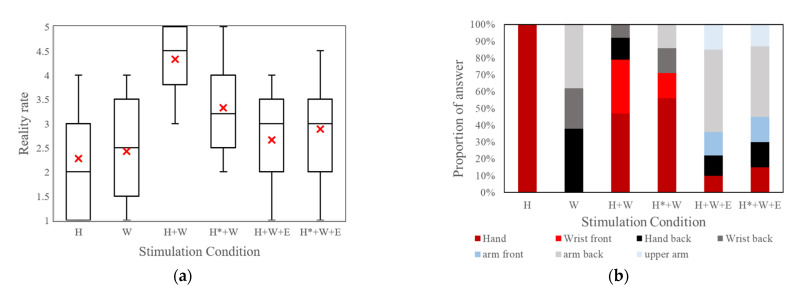
Experimental results. (**a**) Rate of the sensation reality (the red crosses represent the mean of the rating value) and (**b**) Position where the “contact” sensation was sensed.

**Figure 5 sensors-23-00431-f005:**
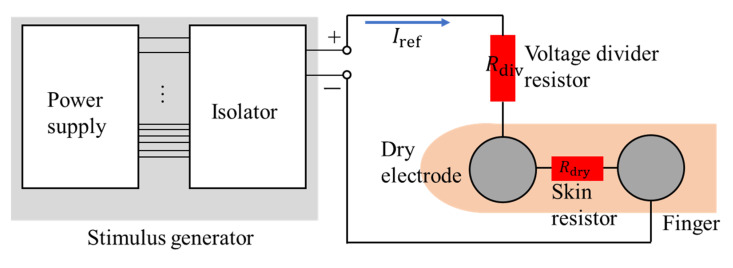
Voltage divider circuit.

**Figure 6 sensors-23-00431-f006:**
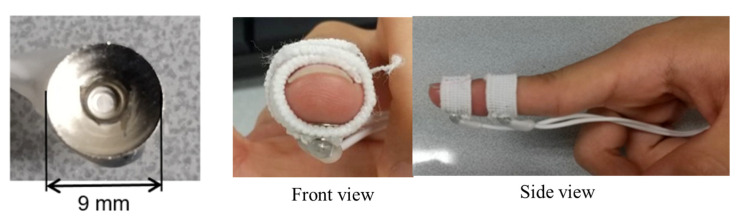
Dry electrode for finger stimulation and its installation.

**Figure 7 sensors-23-00431-f007:**
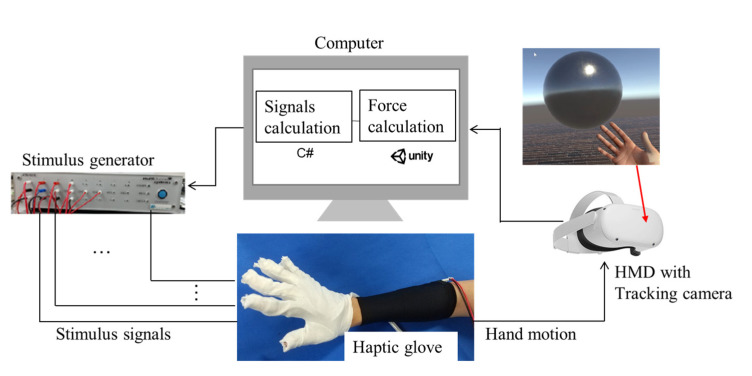
Developed wearable haptic glove system.

**Figure 8 sensors-23-00431-f008:**
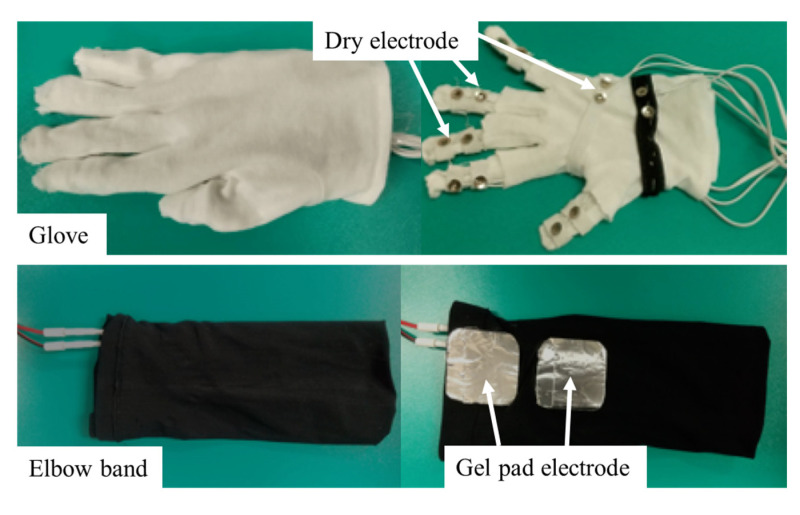
The electrode layout inside the haptic glove.

**Figure 9 sensors-23-00431-f009:**
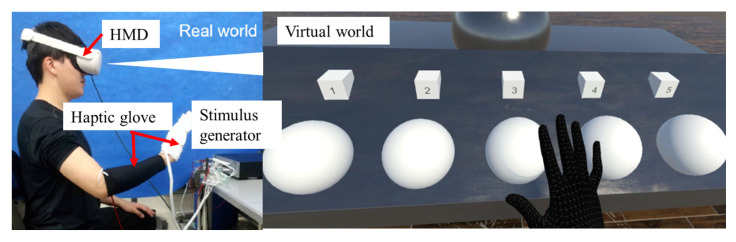
The appearance of the experiment and a view of the virtual world.

**Figure 10 sensors-23-00431-f010:**
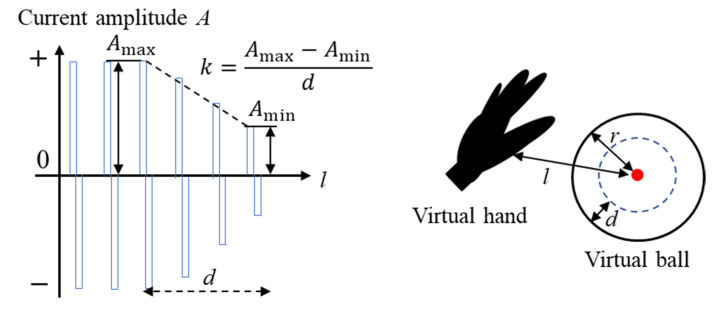
Schematic of determining the stimulus amplitude.

**Figure 11 sensors-23-00431-f011:**
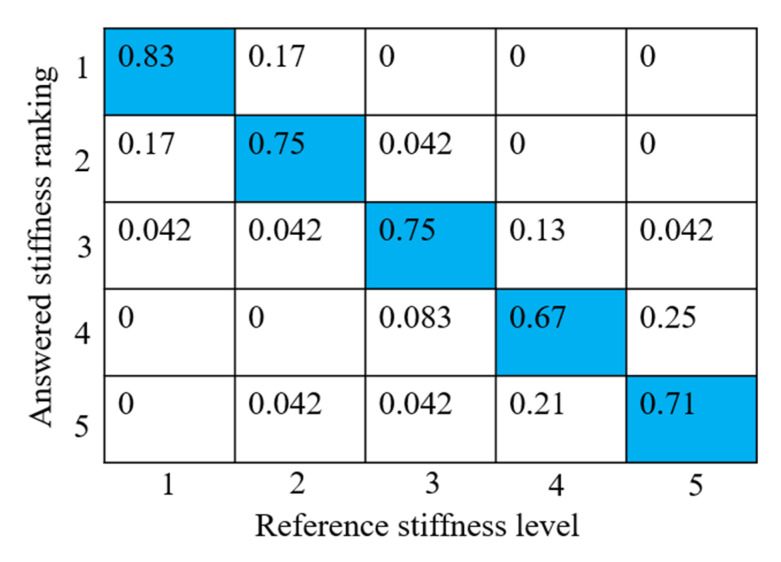
Results of the stiffness discrimination experiment.

**Table 1 sensors-23-00431-t001:** Parameters of the stimulus signals to each subject.

Subject no.	Stimulating Area	$A_{\mathrm{impact}}\left(\mathrm{mA}\right)$	$A_{\mathrm{touch}}\left(\mathrm{mA}\right)$	$A_{\mathrm{con}}\left(\mathrm{mA}\right)$	$A_{\mathrm{int}}\left(\mathrm{mA}\right)$
1	Hand palm (P)	−	−	6.0	7.0
	Wrist extensor digitorum (W)	11.0	9.0	−	−
	Elbow biceps (E)	11.0	10.0	−	−
2	Hand palm (P)	−	−	6.0	9.0
	Wrist extensor digitorum (W)	13.0	14.0	−	−
	Elbow biceps (E)	14.0	12.0	−	−
3	Hand palm (P)	−	−	4.0	7.0
	Wrist extensor digitorum (W)	8.0	6.5	−	−
	Elbow biceps (E)	11.0	9.0	−	−
4	Hand palm (P)	−	−	6.0	8.0
	Wrist extensor digitorum (W)	11.0	9.0	−	−
	Elbow biceps (E)	11.0	11.0	−	−
5	Hand palm (P)	−	−	6.0	8.0
	Wrist extensor digitorum (W)	11.0	9.0	−	−
	Elbow biceps (E)	12.0	11.0	−	−
6	Hand palm (P)	−	−	5.0	8.0
	Wrist extensor digitorum (W)	12.0	9.0	−	−
	Elbow biceps (E)	11.0	10.0	−	−
7	Hand palm (P)	−	−	4.0	5.0
	Wrist extensor digitorum (W)	9.0	7.0	−	−
	Elbow biceps (E)	12.0	10.0	−	−
8	Hand palm (P)	−	−	8.0	4.0
	Wrist extensor digitorum (W)	8.0	7.5	−	−
	Elbow biceps (E)	11.0	11.0	−	−

**Table 2 sensors-23-00431-t002:** Experimental conditions.

Condition	Stimulating Area	Stimulus Signals
H	Hand palm only	Intermittent
W	Wrist extensor digitorum only	Changing amplitude
H + W	Palm, wrist extensor digitorum	Intermittent + change amplitude
H* + W		Continuous + change amplitude
H + W + E	Palm, wrist extensor digitorum, elbow biceps	Intermittent + change amplitude
H* + W + E		Continuous + change amplitude

**Table 3 sensors-23-00431-t003:** Parameters of the stimulus signals to each subject.

Stimulus Area	Stimulus Signals	Pulse Width (µs)	Frequency (Hz)	Current Amplitude (mA)	Duration (s)
DIP and PIP joints of five fingers	Intermittent bidirectional square wave (Figure 2b)	200	50	2.4	0.4

## Data Availability

Data sharing not applicable.

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
