# Peer review of "Development of a Wearable Haptic Glove Presenting Haptic Sensation by Electrical Stimulation"

_sensors, 2022, doi:10.3390/s23010431_

Round 1
Reviewer 1 Report
Review
In this paper, authors present a novel method to generate tactile feedback using electric signals only. On the way, sensation areas on the forearm, eliminating tingling sensation is maintained. Finally, a lightweight prototype glove was developed to test the signals by identifying different ball stiffness in the virtual world.
The subject is interesting, basically well presented. However, some questions and remarks should be addressed.
My recommendation is acceptance after revision.
Comments
Lines 31-32:
"The restricted movement range of these devices, however, limits the range of operational motion."
What do you mean by restricted range? Users can not make large scale movements? If so, it is because of the actuators? Are they too big and heavy? Really small actuators can be used. We recently applied small AC motors in button-size on the arm and its weight was not limiting (the wiring was). This issue can be very specific, as a glove may look completely different with larger actuators than a vest or armband. I wouldn't be so general to state every actuator-based solution to be (too) heavy.
It is not clear why the pure electrical stimuli is better than mixed stimuli (if it is because of the mass difference, please specify).
Line 33: Gyper Grasp? It is CyberGrasp.
Line 40: "widely used in the treatment and rehabilitation of physical paralysis". This is true, maybe some recent references (review article?) could be added.
Line 54: snake baiting? you mean snake biting?
Lines 72-73:
"The second reason is the tingling sensation caused from the instability of the stimulus signals."
Is this really the cause of tingling sensation? I don't know, I'm asking. Small currents on the skin, especially if wet (licking the electrodes of a battery), cause the feeling of tingling. It is DC, but I assume AC would do that too. I wonder how "stable" this should be, so maybe a reference here would also do good. This is important to clarify, because one of your outcomes is to reduce/avoid this by stabilizing the current.
"The presentation of haptic sensations on the palm and the fingertips is necessary for sensation reality"
This also needs reference. You mean simultaneous presentation? Or either of the body parts? Sensation reality on the arm? Because we don't need palm and fingertips to feel haptics on other body parts. Please clarify.
Line 76: "The amplified of the stimulus" - You mean the amplification? Amplified stimulus?
Line 108: 30 mm thickness? Is the Greek letter "fi" the best to describe this dimension?
Line 115:
"by applying an electrical stimulus to the biceps and extensor digitorum muscles, which are responsible for wrist and elbow motions, respectively. "
Is that not the other way round? Biceps for elbow motion and ext.dig. for wrist?
Figure 1:
I understand that ext.dig.muscle is responsible for wrist motion and if triggered, this is a good excitation point. However, this figure is strange, as you show "wrist" that is clearly the elbow and forearm, and "elbow" that is the upper arm (biceps). Until this point, I thought "wrist" means the part right above the hand. You use elbow and wrist position throughout the paper based on Fig.1.
Furthermore, you use "forearm" in Line 176, which seems to be more correct. This is confusing.
Line 134:
"The frequency of all stimulus signals in Figure 2 is 50 Hz, the pulse width is 200 µs"
Why 50 Hz and 200 usec? No problem with these values, but there could be some explanation: is this based on former experiments?
In discussion section there is explanation about using 50 Hz, maybe citations would be good earlier why did you prefer these. In Line 328 you say "we compared the stimulus signals used in other similar studies.” – Reference?
Line 163:
" Figure 4a shows the reality rated by the subject rate" - This sentence is strange.
Line 223:
You use "PC" and you already used PC in Line 131. It is clearly a different thing, here you mean personal computer, maybe this time it would be better to avoid the PC abbreviation.
Lin 229:
Again, the word used for hand and wrist is still confusing.
" When the user wears the elbow band, two gel pad electrodes inside the black elbow band will contact the extensor digitorum muscle" - this is correct! But this is not the wrist.
Lines 252-254:
In the glove a different electrode was used instead of the gel electrodes presented before. The gel electrodes were only used for determining the positions on the body?
A general comment: Although only 8 subjects tested the gloves, it would be interesting to know, if your original goal to create a glove-system using electric signals only was successful in terms of comfort, user friendliness, ergonomy etc.
Is this with the “elbow band” and all together a better solution than former attempts with mixed signals? If you don’t know yet, do you plan to make a comparison and/or informal query?
Author Response
Dear reviewer:
Thank you for your constructive recommendations that helped me improve the quality of the manuscript. I have revised the manuscript according to your recommendations.
Sincerely,
Dongbo Zhou

Reviewer 2 Report
This paper investigated the impact of electrode position and signal waveforms for eliciting force feedback during hand touch. The authors resolved technical challenges, including the instability of electrotactile current and wearability of the haptic feedback device, through their device design. Most part of the paper is written clearly. The introduction has a proper summary of relevant works. Most results are presented clearly. The discussion is very informative and can inspire many future investigations about electrotactile feedback. However, I still have some concerns for the authors to address.
In the first paragraph of the introduction section, I feel it would be valuable to also cite recent works about haptic feedback utilizing functional materials, such as dielectric actuators and shape memory alloys. As the weight and form factor of the functional materials is quite competitive with the electrotactile feedback method described in this paper.
line 42, a citation is desired for the claim that the users will feel external forces due to the stimulation of the antagonistic muscle
I cannot follow the logic connecting the sentences in lines 72-74. A better logic flow is desired.
line 141, the intention for studying the continuous and intermittent signals is unclear. Why does the design of intermittent signals provide no feedback during the touch phase? is there a reason behind this? can a reference be provided here?
line 204, the diameter and contact area of the dry electrode do not agree with each other, there could be a typo or a calculation mistake.
line 231, why use a gel-pad electrode for the elbow and a dry electrode for the hand? A detailed reason is desired. The sentence in line 252 further confused me: Will the results in 2.2.1. change when the configuration switched from dry to gel-pad electrode?
line 264, how did you measure the 5N force? A detailed explanation is appreciated.
line 340, a reference is desirable for the claim "just noticeable difference to distinguish difference increases as the reference increases".
Figure 4a should explain the meaning of the red cross in its caption
Overall, I find this paper quite interesting and valuable for the audience studying haptic feedback. However, my biggest concern is that this study is very much focused on feedback, not sensing. Thus, the topic of this research may not align well with the scope of the Sensors journal. I assume this work is more suited to other journals.
Author Response

(The authors gave the same response as above.)
